# Cancer and Immune Checkpoint Inhibitor Treatment in the Era of SARS-CoV-2 Infection

**DOI:** 10.3390/cancers12113383

**Published:** 2020-11-16

**Authors:** Thilo Gambichler, Judith Reuther, Christina H. Scheel, Laura Susok, Peter Kern, Jürgen C. Becker

**Affiliations:** 1Department of Dermatology, Skin Cancer Center, Ruhr-University Bochum, 44791 Bochum, Germany; judith.reuther@klinikum-bochum.de (J.R.); christina.scheel@klinikum-bochum.de (C.H.S.); laura.susok@klinikum-bochum.de (L.S.); 2Institute of Stem Cell Research, Helmholtz Center Munich, 85764 Neuherberg, Germany; 3Department of Gynecology, St. Elisabeth-Hospital, Ruhr University Bochum, 44787 Bochum, Germany; peter.kern@klinikum-bochum.de; 4Translational Skin Cancer Research, German Cancer Consortium (DKTK), Dermatology, University Duisburg-Essen, 45141 Essen, Germany; j.becker@dkfz-heidelberg.de; 5German Cancer Research Center (DKFZ), 69120 Heidelberg, Germany

**Keywords:** immune checkpoint inhibitors, programmed cell death protein, cytotoxic T lymphocyte antigen-4, COVID-19, coronavirus

## Abstract

**Simple Summary:**

The introduction of immune checkpoint inhibitors (ICI) in 2011 revolutionized the management of many solid cancers and hematological malignancies. However, there are concerns regarding the use of ICI in the era of COVID-19. We present currently available information on the pros and cons of using ICI in cancer patients with respect to the risk of acquiring an infection by SARS-CoV2 and mortality from COVID-19. By means of the present paper, clinicians and researchers may update their knowledge on a highly topical clinical question—is the use of ICI in cancer patients with SARS-CoV2 infection harmful with respect to COVID-19 outcome?

**Abstract:**

Whether cancer patients receiving immune checkpoint inhibitors (ICI) are at an increased risk of severe infection and mortality during the corona pandemic is a hotly debated topic that will continue to evolve. Here, we summarize and discuss current studies regarding COVID-19 and anti-cancer treatment with an emphasis on ICI. Importantly, several lines of evidence suggest that patients currently treated with ICI do not display an increased vulnerability to infection with SARS-CoV-2. Data regarding morbidity and mortality associated with COVID-19 in cancer patients receiving ICI are less clear and often conflicting. Although mostly based on experimental data, it is possible that ICI can promote the exacerbated immune response associated with adverse outcome in COVID-19 patients. On the other hand, mounting evidence suggests that ICI might even be useful in the treatment of viral infections by preventing or ameliorating T cell exhaustion. In this context, the right timing of treatment might be essential. Nevertheless, some cancer patients treated with ICI experience autoimmune-related side effects that require the use of immunosuppressive therapies, which in turn may promote a severe course of infection with SARS-CoV-2. Although there is clear evidence that withholding ICI will have more serious consequences, further studies are urgently needed in to better evaluate the effects of ICI in patients with COVID-19 and the use of ICI during the corona pandemic in general.

## 1. Introduction

The coronavirus disease 2019 (COVID-19) outbreak, caused by the severe acute respiratory syndrome coronavirus 2 (SARS-CoV-2), was declared a pandemic by the WHO in March 2020. Since its emergence, 33,995,564 cases of COVID-19 have been confirmed with 1,014,557 deaths reported worldwide up through the 1 October 2020 [1]. Death in severe cases is associated with viral pneumonia, while common symptoms of COVID-19 include high fever, dry cough, fatigue and gastrointestinal issues such as vomiting and diarrhea. The case fatality rate (CFR) has been calculated to be as high as 1%. This is much higher than, for example, the CFR typically determined for seasonal influenza (app. 0.1%). Older age (particularly older than 70 years), chronic respiratory disease, diabetes, obesity, cardiovascular disease and—probably—both cancer and its respective therapies, all present risk factors for the development of severe course of COVID-19 eventually resulting in death [2,3,4]. Of note, most of these published data originate from China, Italy, France, and the USA. Thus, some of these observations may be population-specific and therefore may not be completely comparable with COVID-19 outbreaks in other countries.

Immune checkpoint inhibitors (ICI) serve to activate an anti-tumor response by blocking co-inhibitory pathways that result in immune-mediated killing of tumor cells. Thus, the introduction of ICI in 2011 revolutionized the management of many solid cancers and hematological malignancies (e.g., colorectal cancer, Hodgkin lymphoma, lung cancer, melanoma, Merkel cell carcinoma, renal carcinoma, urothelial cancer) [5,6,7,8]. In this update of our previous review [9], we present currently available information on the pros and cons of using ICI in cancer patients with respect to the risk of acquiring an infection by SARS-CoV2 and mortality from COVID-19.

## 2. Mode of Action of ICI

Immune checkpoints consist of both co-stimulators as well as co-inhibitors that together regulate T lymphocyte activation [10]. The activation of co-stimulatory receptors such as CD28 together with triggering the T-cell receptor by the specifically recognized antigen-major histocompatibility complex (MHC) allows T lymphocyte proliferation, differentiation, and migration. If the target cell, however, expresses the above-mentioned inhibitory immune checkpoints, T-cell activation will be suppressed. Expression of programmed cell death protein ligand-1 (PD-L1) is widespread and includes several different types of immune as well as other cells, including tumor and some epithelial cells. The interaction of PD-1 with its ligands inhibits signaling by an activated TCR and thus, results in inhibition of T cell proliferation and survival, cytokine secretion, induces apoptosis of tumor-infiltrating lymphocytes, and promotes the differentiation of CD4+ lymphocytes into T regulatory cells (Treg) [5,6,7,10,11]. Importantly, expression of PD-L1 has been observed in many cancers, for example in response to interferon signaling, thereby limiting anti-cancer immune responses and promoting escape from immune surveillance.

Expression of CTLA-4 is induced upon T cell activation and cannot be found on naïve T lymphocytes. CTLA-4 binds to B7 proteins, competing directly with CD28 as a crucial co-stimulatory signal. Indeed, the ratio between CTLA-4 and CD28 binding to B7 determines T cell anergy versus activation. Moreover, CTLA-4 controls the amplitude of T cell activation during the early priming phase in lymphoid organs and represents an important mechanism in the prevention of excessive immune responses by stopping autoreactive T lymphocytes at the initial stages [5,6,7,8,10,11].

## 3. ICI, Viral Infection, and T Cell Exhaustion

Many diseases caused by chronic viral infections such as hepatitis B and C virus (HBV, HCV), HIV, and JC-virus (causing progressive multifocal leukoencephalopathy) are characterized by T cell exhaustion [9]. After being first described in the context of chronic viral infections, exhausted T lymphocytes with a similar phenotype were also detected in the microenvironment of tumors, suggesting that T cell exhaustion is an important factor in an ineffective, chronic immune response against cancer as well as infectious agents [6,12,13,14]. Altered expression of pro-inflammatory cytokines such as loss of interleukin 2 (IL-2) production, impaired proliferation, and diminished cytotoxicity all represent hallmarks of exhausted T lymphocytes [6,12,13,14]. Importantly, overexpression of the immune checkpoint receptor PD-1 is another characteristic. Consequently, the weakening of the immune response observed during chronic infections as well as the development of different malignancies share similarities at the functional and molecular level that together indicate that therapy with ICI should not be harmful to tumor patients with concomitant viral infection and may even provide some benefit. Nonetheless, such patients are currently excluded from many treatment protocols with ICI [13,14].

Although no increased risk for the acquisition of infections during treatment with ICI was observed in clinical studies [5,6,7,8,13,14], the most common side effects of ICI may caution against the use of ICI in patients with preexisting chronic viral infections. These include a variety of autoimmune reactions including thyreoiditis, pneumonitis, colitis, hypophysitis, hepatitis, and others [15,16,17]. Such immune-related adverse events (irAEs) often require immunosuppressive therapies, for example using high-dose corticosteroids and/or TNF-α blockers. This, in turn, carries the risk of exacerbating a latent or chronic viral infection as well as predisposing for newly acquired infections as a secondary consequence of ICI therapy. In this context, it is particularly important to note that the morbidity of chronic viral disease is caused by collateral damage due to chronic inflammation that results from failure of viral clearance, which may be ameliorated by treatment with ICI.

With respect to COVID-19, some serological factors predict severe disease course. These include increased levels of proinflammatory cytokines IL-6, IL-10 and tumor necrosis factor-α (TNF-α) as well as elevated lactate dehydrogenase, pro-calcitonin, and D-dimers [3,4,18]. Moreover, decreased lymphocyte counts, particularly NK and CD8+ cells, appear to be negatively associated with patient survival. A decrease in the total number of T lymphocytes in peripheral blood was particularly pronounced in older patients prone to severe disease course and generally in patients requiring admission to the Intensive Care Unit (ICU) [18]. Conversely, in patients that recovered, T cell counts as well as cytokine levels normalized [18]. Moreover, expression of PD-1 and T cell immunoglobulin and mucin domain-containing protein 3 (Tim-3)—another inhibitory checkpoint protein recently characterized to play a role in T cell exhaustion—were found to be increased compared to healthy controls [18]. Pronounced Tim-3 expression was discovered on the cell surface of T cells by flow cytometry as patients became symptomatic. Together, these data suggest that T cell exhaustion, which also leads to apoptosis of T cells, might contribute to the lymphopenia observed in COVID-10 patients with severe symptoms [18,19].

## 4. Clinical Outcome of ICI-Treated Cancer Patients with SARS-CoV-2 Infection

To date, limited and contradictory data are available addressing the clinical outcome of patients receiving anti-cancer treatment upon SARS-CoV2 infection [20,21,22,23,24,25]. It must be stressed that given the relatively recent emergence of COVID-19, many studies published as of yet, necessarily present rather preliminary data. For example, in some cases, COVID-19 was not confirmed by RT-PCR, disease stage was not addressed, and treatment modalities were not analyzed in detail. Therefore, it is no surprise that outcomes differed between studies. Other factors hampering reproducibility include small sample size and retrospective study design. Moreover, the prevalence of cancer among COVID-19 patients was not consistently compared to that of cancer patients in the community, and there were uncertainties whether the patient died from COVID-19-related complications, treatment/diagnostic delays caused by the pandemic, or other causes [25]. So far, the reported frequency of cancer patients also suffering from COVID-19 ranges from 0.5 to 6%; notably, these numbers likely depend on the test strategies applied and the number of cancer patients surveyed in different evaluations [25,26]. Saini et al. [27] recently reported that patients with cancer who develop COVID-19 have a high probability of mortality [25.6% (95% CI: 22.0–29.5%)]. A very frequently cited paper studied 18 cancer patients with COVID-19 [28]; this study concluded that cancer patients were both at a higher risk for SARS-CoV-2 infection as well as for a severe or fatal outcome of COVID-19. However, only 4 of the 18 patients (almost 30% lung cancer) received anti-cancer treatment [chemotherapy (*n* = 2), immunotherapy (*n* = 1), surgery (*n* = 1)] within the past month prior to COVID-19 diagnosis, whereas 12 patients were cancer survivors in routine follow-up after primary resection (unknown treatment status in 2 patients) [28]. Zhang et al. [29] examined 28 cancer patients with COVID-19, among them 25% with lung cancer. They observed that cancer patients suffered from deteriorating condition and poor outcome from COVID-19, in particular if anti-cancer treatment was received within the last 14 days before the diagnosis of COVID-19. Six of these patients received targeted therapies, another six received ICI. Notably, COVID-19 outcome was not associated with a specific anti-cancer treatment [29]. Miyashita et al. [30] observed an increased risk for becoming dependent on mechanical ventilation in 334 cancer patients with COVID-19 compared to non-cancer patients at the age of 66–80 years. Interestingly, cancer patients younger than 50 years exhibited a significantly higher mortality rate from COVID-19. However, the impact of treatment modality and stage of disease were not specifically addressed [30]. Yang et al. [31] studied 205 cancer patients with COVID-19 that were confirmed by PCR. They observed high mortality and listed several unfavorable prognostic factors including receiving chemotherapy within 4 weeks before COVID-19 onset as well as male sex [31]. ICI was prescribed only in seven cases, among those, four patients survived and three died from COVID19-related complications. In a multicenter study, 105 cancer patients with COVID-19 were compared to age-matched non-cancer patients with confirmed COVID-19 [32]. Among these, patients with hematologic malignancies, lung cancer, or stage IV disease exhibited the highest frequency of severe events. Among these, six lung cancer patients received ICI, of whom two died [32].

After correction for age and sex, Lee et al. [33] also observed that patients with hematological malignancies, who had recently received chemotherapy, displayed an increased risk of death during COVID-19-associated hospital admission. Mehta et al. [34] also reported a significant risk for patients (*n* = 218) with cancer infected by COVID-19, with a significant increase in mortality. Again, the highest susceptibility for an increased risk appeared to occur in patients with hematologic and lung malignancies [34]. In this study, only four patients received ICI [34]. Taken together, these studies suggest a higher risk for a severe course of COVID-19 in cancer patients, which also appears to apply to those cancer patients that received ICI, even though sample sizes were very limited.

By contrast, some studies report that cancer patients are not at any increased risk for a more severe course of COVID-19 or increased mortality. Minotti et al. [35] report that immunosuppressed patients (children as well adults) with COVID-19 (*n* = 110; cancer, post-transplantation, immunodeficiency) appear to have a more favorable outcome compared to the general population. The authors hypothesized a protective role of a weakened immune response with respect to COVID-19-related complications. Indeed, the latter are often caused by an overshooting immune response [35]. Moreover, Bi et al. [36] reported that asymptomatic infections with SARS-CoV-2 are more likely to occur in cancer patients as compared to non-cancer-afflicted caregivers located in a similar environment of exposure.

Finally, some studies specifically focused on cancer patients with COVID-19 who received ICI. Wu et al. [26] reported on 11 patients (six of them suffering from lung cancer) who received ICI shortly prior to or while contracting COVID-19: about two-thirds developed severe complications. Moreover, it was speculated that severity and mortality due to COVID-19 could be linked with the number of ICI cycles received [26]. Robilotti et al. [37] reported on 423 cancer patients suffering from COVID-19, of which 31 were treated with ICI. They reported that age above 65 years and ICI treatment were risk factors for severe disease requiring hospitalization with a CFR of 12%. In contrast to other studies, chemotherapy and major surgery were not determined as significant risk factors. The Robilotti study [37] also found that the ICI-treated patients experienced worse COVID-19 outcome without the confounding effect of lung cancer [37,38,39]. Importantly, the authors introduced an endpoint based on significant oxygen need, which was more common than death [35]. Lee et al. [40], on the other hand, reported that mortality from COVID-19 in cancer patients appears to be principally driven by age, gender, and comorbidities. They were not able to identify evidence that cancer patients on cytotoxic chemotherapy or other anticancer treatment, including 44 patients receiving ICI, were at an increased risk of mortality from COVID-19 compared to those not on active treatment.

Luo et al. [41] reported that the severity of RT-PCR-confirmed COVID-19 in 66 patients with lung cancer was high, including the need for hospitalization in more than half of patients and a CFR of almost 25%. After adjustment for smoking status prior, however, ICI exposure was not associated with an increased risk for severe disease. Hence, the authors concluded that PD-1 blockade does not appear to affect the severity of COVID-19 [41]. Quaglino et al. [42] recently reported their data on 80 melanoma patients who received ICI therapy during the pandemic. Their experience supports the possibility of continuing ICI in metastatic melanoma patients, even if evaluating on a patient basis (elderly, comorbidities, ongoing response, adjuvant treatment) the possibility of delaying the subsequent course [42]. Moreover, several case series and case reports did not describe any detrimental effects of using ICI in cancer patients with SARS-CoV-2 infection [43,44,45]. Gonzalez-Cao et al. [46] observed that the risk of death in melanoma patients undergoing treatment with ICI does not exceed the global risk of death in this population. Interestingly, Pala et al. [47] recently reported that the incidence of symptomatic COVID-19 infection observed in their cohort of patients with advanced malignant melanoma treated with immunotherapy was meaningfully lower as compared with that reported in the overall population in Italy as well as in patients affected by other solid tumors. Finally, a recent Italian survey on ICI treatment in different cancer types revealed that the frequency of ICI treatment was not drastically reduced during the pandemic [48].

## 5. ICI, Cytokine Release Syndrome, Autoimmunity, and COVID-19

An important aspect determining the severity of COVID-19 course is the so-called “cytokine storm”. At least three inflammatory disorders or syndromes associated with immune dysfunction have been described in the context of cellular therapies and immunotherapies, including cytokine release syndrome (CRS), immune reconstitution (inflammatory) syndrome (IRIS), and secondary hemophagocytic lymphohistiocytosis, and all have clinical and laboratory characteristics in common with COVID19 [49,50,51,52]. Importantly, IRIS can manifest systemically or localize to the lung as acute respiratory distress syndrome [49,50]. CRS describes a phenomenon of massive, exuberant inflammatory reaction resulting in an excess of proinflammatory cytokines such as IL-6, IL-10, and TNF-α induced by different stimuli including infectious agents [18,53,54,55]. Importantly, SARS-CoV-2 has been associated with increased amounts of proinflammatory cytokines detectable in peripheral blood, which are suspected to cause pulmonary inflammation and extensive lung damage [56]. Moreover, patients with severe COVID-19 also show changes in the lung stroma that suggest the development of lung fibrosis at later time points [57]. Mechanistically, SARS-CoV-2 infection appears to be associated with the activation of both T helper 1 (Th-1) and Th-2 lymphocytes. Due to the cross-reactivity with viral proteins, particularly spike surface proteins with host epitopes, SARS-CoV-2 may also pose the risk of autoimmunity, as has been previously described for MERS-CoV and SARS-CoV [57].

In animal experiments involving SARS-CoV and MERS-CoV, severe Th-2-driven autoimmune reactions were reported, particularly upon challenge following vaccination [57,58]. In light of these data, administration of ICI in COVID-19 patients could indeed pose a risk for immune over-activation and/or aggravation of autoimmune processes [24,59,60,61]. Corroborating these concerns, CRS has also been observed as an irAE in patients receiving ICI in a few rare cases [53,54,62,63]. Moreover, both clinical symptoms as well as radiographic findings for COVID-19-induced pneumonitis are very similar to those found in irAE-induced pneumonitis, which may render appropriate diagnosis difficult and complicate appropriate patient management [61,64,65,66]. In patients receiving anti-PD-1-antibodies, autoimmune-induced pneumonitis may occur in as many as 2.7% cases; in patients receiving combined PD-1- and CTLA-4-blockade, the frequency may be as high as 6.6% [21,63,67]. Patients with non-small cell lung cancer even may develop autoimmune pneumonitis in up to 20% of cases upon ICI [67]. Systemic corticosteroids and other immunosuppressive agents such as mycophenolate acid or TNF-α blockers are the standard of care for ICI-associated irAEs. Whether the use of systemic corticosteroids is harmful in the setting of COVID-19 is much clearer today than it was 6 months ago. In a prospective meta-analysis of clinical trials of critically ill COVID-19 patients, administration of systemic corticosteroids was associated with significantly reduced 28 day all-cause mortality [68]. The Writing Committee for the REMAP-CAP Investigators states [69] that early and short-time use of low-dose corticosteroids likely constitutes a feasible approach to manage SARS-CoV-2- related pneumonitis [68,70]. Hence, ICI-induced irAEs and COVID-19-associated pneumonitis are both safely managed with corticosteroids.

Moreover, based on the frequent occurrence of CRS, which includes TNF-α as one of the predominant proinflammatory cytokines, during severe courses of COVID-19, it was speculated that TNF-α blockers such as infliximab may be an effective treatment of CRS [71]. In this context, antibodies against the proinflammatory cytokine IL-6 (e.g., tocilizumab) are currently under intense investigation for use in COVID-19 patients. Tocilizumab is already approved in the USA to manage CRS occurring as a treatment for irAE caused by chimeric antigen receptor (CAR) T cells [72,73]. Additionally, it has been demonstrated that tocilizumab is effective in the management of ICI-associated irAEs not sufficiently controlled by corticosteroids [72]. Furthermore, an ongoing phase II trial investigates a combination of tocilizumab with anti-PD-1/CTLA-4 therapy in order to diminish irAEs in melanoma (NCT03999749) [71]. In conclusion, several cytokine-blockers (anti-IL-6, anti-IL-1, anti-TNF-α etc.) are currently under investigation for COVID-19 patients that are already in use or currently tested for the treatment of irAE [54].

## 6. ICI—A Potential Strategy against SARS-CoV-2 Infection?

Importantly, in contrast to chemotherapeutic regimens, ICI cannot be considered to induce immunosuppression. In a meta-analysis comprising 9324 patients, the frequency of neutropenia was less than 1%, and lymphopenia—a much more significant issue in the context of SARS-CoV-2 infections—is also uncommon under ICI treatment [74,75,76]. Nevertheless, lymphopenia needs to be regarded as a general hallmark of metastatic cancer patients. Several mechanisms may account for this finding: (I) killing of lymphocytes by tumor cells expressing proapoptotic ligands, (II) production of immunosuppressive cyto/chemokines by both tumor and tumor stroma cells, (III) presence of CTLA-4-expressing Tregs, (IV) activation-induced cell death of lymphocytes, and (V) immunosuppressive anti-cancer treatments [77,78]. Moreover, the expression of immune checkpoint proteins (PD-1, Tim-3, CTLA-4) by tumor-infiltrating lymphocytes known to be involved in the inhibition of T lymphocyte activation/proliferation could also be a factor in causing peripheral lymphopenia in cancer patients. Consequently, the therapeutic activity of ICI may, in part, lie in the restoration of normal numbers of circulating lymphocytes [77,78].

Previously reported reactivation of viral infections (i.e., cytomegalovirus, HBV) under ICI therapy was mostly observed following immunosuppressive treatment of irAEs [9,79,80]. Shah et al. [80] recently concluded that their data support not excluding cancer patients with concomitant viral infections (e.g., HIV, HBV, HCV) from ICI-based clinical trials or treatment. Hence, it does not appear reasonable to assume that patients undergoing ICI are at higher risk of becoming infected by SARS-CoV-2 or other infectious agents compared to patients without ICI treatment [22]. Additionally, blockade of PD-1 on CD8+ lymphocytes by ICI treatment might be beneficial in COVID-19 patients in order to overcome functional T cell exhaustion and restore vigorous T lymphocytic cytotoxicity against both tumor as well as virally infected cells. However, as discussed by Chiappelli et al. [21], this strategy may only be effective in the initial and intermediate stage of COVID-19 when PD-1 expression on cytotoxic T cells ranges between low and medium levels. At more advanced stages when PD-1 expression is high on CD8+ T lymphocytes, T lymphocyte exhaustion is considered irreversible and thus, ICI may no longer have an effect [21].

Hence, it has been proposed by many authors that ICI may represent an effective approach in the management of COVID-19 patients without cancer [18,19,22,62,81]. A combination of ICI with an anti-IL-6 antibody represents an attractive approach to reduce the risks for both irAEs and CRS frequently observed in severe COVID-19 cases [82,83]. Indeed, a randomized, controlled, open-label, phase II trial to evaluate the efficacy and safety of tocilizumab combined with pembrolizumab (MK-3475) in patients with COVID-19-pneumonia has been registered at ClinicalTrials.gov (NCT04335305; COPERNICO). In another, not yet recruiting trial, the efficiency and security of nivolumab therapy will be investigated in obese individuals with COVID-19 (NCT04413838).

A previous study following this strategy has been suspended as it included a chloroquine analog. Importantly, hydroxychloroquine use has been reported to be ineffective in COVID-19 or even may be associated with higher mortality. Furthermore, two protocols have been registered at ClinicalTrials.gov investigating ICI in COVID-19 patients without cancer: (I) In a phase 2 randomized trial, the protocol CORIMUNO19-NIVO will evaluate the efficacy and safety of nivolumab alone versus standard of care in patients hospitalized in an ICU (NCT04343144) and (II) in an open-label, controlled, single-center pilot study, nivolumab will be employed in adult patients with COVID-19 aiming to investigate the efficacy and safety of nivolumab in relation to viral clearance (NCT04356508).

Finally, it is important to note that a significant portion (up to 14%) of patients with a severe course of COVID-19 might not respond to ICI due to the infection-triggered development of neutralizing antibodies against type I interferons [84] or because they harbor rare mutations in genes required for an efficient interferon response [85]. Importantly, these new data also suggest that cancer patients who develop auto-antibodies during a severe course of COVID-19 might no longer respond to ICI efficiently.

## 7. Conclusions

Our current knowledge on the reciprocal impact of cancer and COVID-19 is limited and unbalanced with respect to the reported cancer entities. ICI was initially developed in and is routinely used as first-line therapy of skin cancer (e.g., melanoma, Merkel cell carcinoma, cutaneous squamous cell carcinoma); there is now increasing data regarding the outcomes of melanoma patients once ICI therapy and COVID-19 coincide [42,45,46,47,86]. However, the experience in lung cancer, breast cancer, gastrointestinal tumors, bladder cancer, and hematological malignancies are most frequently reported. Here, it appears that patients with lung cancer have a rather severe course of COVID-19, presumably caused by preexisting lung damage by both prior therapeutic interventions (e.g., surgery) and a frequent history of smoking [32,34,87].

Based on our comprehensive review of the current literature, we conclude that anti-PD-1/PD-L1- and/or anti-CTLA-4-based cancer treatment does not appear to increase vulnerability for SARS-CoV-2 infection per se. However, whether the use of ICI in cancer patients with COVID-19 increases morbidity and mortality related to the SARS-CoV-2 infection cannot be definitively answered at this time. From a functional point of view, considering the mode of action of ICI, this approach appears not harmful for cancer patients with concomitant SARS-CoV-2 infection [22,83,88,89,90]. Hence, COVID-19 may not be considered as a strict contraindication for patients that are scheduled for or are currently under ICI treatment for malignancies, in particular patients with distant metastatic disease (Table 1) [86,91]. In this regard, the National Institute for Health and Care Excellence in Great Britain has recently updated their NHS England interim treatment options for cancer patients during the COVID-19 pandemic. Based on their recommendations, many cancers should be first-line treated with ICI-monotherapy in order to reduce the risk of chemotherapy-induced hematological side effects [92]. In order to minimize exposure to healthcare workers and other patients, ICI should be administered in prolonged intervals (e.g., nivolumab every 4 weeks, pembrolizumab every 6 weeks) if in-label and indicated (e.g., patient’s compliance) [92,93]. In rare cases, home-based infusion may represent an option to manage cancer patients during this pandemic. Some patients may also consider discontinuing ICI following an extended duration (i.e., following complete response after 1–2 years of ICI) or pausing ICI treatment in regions of high COVID-19 prevalence [93]. With the focus on melanoma management during the pandemic, the European Society for Medical Oncology (ESMO) [94], the National Comprehensive Cancer Network (NCCN) [95], Nahm et al. [96], and Baumann et al. [97] have also published recommendations and guidelines as recently summarized by Elmas et al. [98].

Over the course of ICI treatment, however, supportive immunosuppressive therapies may be required to treat therapy-associated irAEs, which in turn may increase the risk of SARS-CoV-2 infection depending on the immunosuppressive agents used. The use of immunosuppressive interventions in symptomatic COVID-19 patients may be indicated anyway. In this respect, corticosteroids, TNF-α-blockers, and IL-6-blockers should be preferably used over other immunosuppressive agents that may cause severe lymphopenia. Nonetheless, physicians caring should maintain close surveillance for the occurrence of symptoms or signs suggestive of a SARS-CoV-19 infection or worsening of preexisting COVID-19 for patients receiving immunosuppressants for treatment of ICI-induced irAEs [85].

Since there is no convincing evidence that ICI are immunosuppressive per se, avoiding these agents in cancer patients to minimize the risk of SARS-CoV-2 infection would deprive these patients from a highly effective treatment option [22,23]. If the SARS-CoV-2 outbreak further extends, the detriments from the unavailability of high-level oncology care would likely be greater than that of a COVID-19 infection in cancer patients [99]. Indeed, due to the COVID-19 pandemic, epidemiologic extrapolations foresee a 20% increase in cancer mortality, predominantly because of delays in diagnosis and treatment [100]. Thus, the risk of contracting COVID-19 or worsening its course has to be balanced with the risk of cancer progression [22,39,42,62,101,102].

Finally, increasing lines of evidence suggest that ICI not only represent an anti-tumor treatment, but also hold potential in managing SARS-CoV-2 infections. The latter is reflected by current clinical trial activities. Indeed, functional studies suggest T-cell exhaustion as a fundamental mechanism for unfavorable prognosis of both cancer and viral diseases, including SARS-CoV-2.

## Figures and Tables

**Table 1 cancers-12-03383-t001:** General recommendations for the use of immune checkpoint inhibitor (ICI) therapy for cancer patients in the era of COVID-19 pandemic.

1.	ICI should be considered in approved indications, in particular in advanced cancers. In the adjuvant setting or in particular cancers such as lung cancer, one must carefully weigh the pro and cons. In order to reduce toxicity-induced complications, mono-ICI should be preferred, in particular in prolonged intervals if indicated. Combination ICI may still be indicated for patients with very high risk (e.g., brain metastasis).
2.	Patients should be regularly checked regarding clinical signs of COVID-19 and the well-known risk factors (e.g., high-risk countries); adequate testing capacities should be ensured for all symptomatic and/or suspected COVID-19 patients before ICI is initiated; RT-PCR tests for asymptomatic/unsuspected patients may be considered at treatment appointments if testing capacities are solid.
3.	In order to minimize exposure to healthcare workers and other patients, ICI should be administered in prolonged protocols (e.g., nivolumab every 4 weeks, pembrolizumab every 6 weeks) if in label and indicated (e.g., patient’s compliance). In order to timely detect irAE, regular virtual visits may be considered (e.g., twice weekly via phone or other media). If oral targeted therapy is indicated and equivalent, it may be favored over ICI.
4.	In patients with complete response or very low tumor burden, ICI may be withheld (at least in certain cancers); initiation of ICI may be delayed in patients with low pre-treatment tumor burden; this is especially true for confirmed COVID-19 cases until resolution of COVID-19 symptoms.
5.	Critically ill patients (respiratory failure etc.) with known/suspected COVID-19 should receive corticosteroids when ICI-induced pneumonitis, colitis or hepatitis cannot be excluded.

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
