# Peer review of "Cancer and Immune Checkpoint Inhibitor Treatment in the Era of SARS-CoV-2 Infection"

_cancers, 2020, doi:10.3390/cancers12113383_

Round 1

Reviewer 1 Report

minor revisions

This is a timely review, although several other similar reports have also been published in 2020, and all of these reviews are limited by the limited available data. That said, this is perhaps one of the most comprehensive reviews of its type and I have made a few suggestions to ensure that as much of the relevant literature is included. The review is very well written and the tabulated recommendations a useful addition

Comments

The authors could cite studies using ICI in the presence of other viral infections - HIV, HBV and HCV whihc are informative, for instance El-Khoueiry AB, Shah NJ papers

Line 181: This comment seems to be based on comparison of 75 papers and 110 immunosuppressed patients - 98 with cancer, so I don’t think the statement that cancer patients have a more severe course can be made, at least the caveat that few non-cancer patients included should be stated

The Robilotti study [37] also found that the ICI group experienced worse outcome without the confounding effect of lung cancer - and this is an important point that should be highlighted

The Quaglino report recommended that elderly patients cease ICI treatment, and in fact the median age of patients interrupting therapy was 78 years - this will obviously influence the conclusions made, and should be noted for completeness

In section 5, the association of elevated cytokines with ICI-associated toxicity in melanoma should be cited in this section Lim et al.

The following report should also be mentioned Lancet Oncol. 2020;21(7):914-22.

Author Response

We are very grateful for the reviewer comments. We have replied to their comments point by point in italics:

Ref. 1

minor revisions

This is a timely review, although several other similar reports have also been published in 2020, and all of these reviews are limited by the limited available data. That said, this is perhaps one of the most comprehensive reviews of its type and I have made a few suggestions to ensure that as much of the relevant literature is included. The review is very well written and the tabulated recommendations a useful addition

 Comments

The authors could cite studies using ICI in the presence of other viral infections - HIV, HBV and HCV whihc are informative, for instance El-Khoueiry AB, Shah NJ papers

We have included these references in section 6.

El-Khoueiry AB, Sangro B, Yau T, Crocenzi TS, Kudo M, Hsu C, Kim TY, Choo SP, Trojan J, Welling TH Rd, Meyer T, Kang YK, Yeo W, Chopra A, Anderson J, Dela Cruz C, Lang L, Neely J, Tang H, Dastani HB, Melero I. Nivolumab in patients with advanced hepatocellular carcinoma (CheckMate 040): an open-label, non-comparative, phase 1/2 dose escalation and expansion trial. Lancet. 2017 Jun 24;389(10088):2492-2502. doi: 10.1016/S0140-6736(17)31046-2. Epub 2017 Apr 20. PMID: 28434648; PMCID: PMC7539326.

Shah NJ, Al-Shbool G, Blackburn M, Cook M, Belouali A, Liu SV, Madhavan S, He AR, Atkins MB, Gibney GT, Kim C. Safety and efficacy of immune checkpoint inhibitors (ICIs) in cancer patients with HIV, hepatitis B, or hepatitis C viral infection. J Immunother Cancer. 2019 Dec 17;7(1):353. doi: 

Line 181: This comment seems to be based on comparison of 75 papers and 110 immunosuppressed patients - 98 with cancer, so I don’t think the statement that cancer patients have a more severe course can be made, at least the caveat that few non-cancer patients included should be stated

We have deleted this statement.

The Robilotti study [37] also found that the ICI group experienced worse outcome without the confounding effect of lung cancer - and this is an important point that should be highlighted

We have highlighted this statement as requested.

The Quaglino report recommended that elderly patients cease ICI treatment, and in fact the median age of patients interrupting therapy was 78 years - this will obviously influence the conclusions made, and should be noted for completeness

We have more precisely mentioned Quaglino conclusions as stated in the original report.

In section 5, the association of elevated cytokines with ICI-associated toxicity in melanoma should be cited in this section Lim et al.

Unfortunately, we have not found a fitting article for Lim et al.?

The following report should also be mentioned Lancet Oncol. 2020;21(7):914-22.

We have included this reference:

Garassino MC, Whisenant JG, Huang LC, Trama A, Torri V, Agustoni F, Baena J, Banna G, Berardi R, Bettini AC, Bria E, Brighenti M, Cadranel J, De Toma A, Chini C, Cortellini A, Felip E, Finocchiaro G, Garrido P, Genova C, Giusti R, Gregorc V, Grossi F, Grosso F, Intagliata S, La Verde N, Liu SV, Mazieres J, Mercadante E, Michielin O, Minuti G, Moro-Sibilot D, Pasello G, Passaro A, Scotti V, Solli P, Stroppa E, Tiseo M, Viscardi G, Voltolini L, Wu YL, Zai S, Pancaldi V, Dingemans AM, Van Meerbeeck J, Barlesi F, Wakelee H, Peters S, Horn L; TERAVOLT investigators. COVID-19 in patients with thoracic malignancies (TERAVOLT): first results of an international, registry-based, cohort study. Lancet Oncol. 2020 Jul;21(7):914-922. doi: 10.1016/S1470-2045(20)30314-4. Epub 2020 Jun 12. PMID: 32539942; PMCID: PMC7292610.

Reviewer 2 Report

ICI can promote the exacerbated immune response associated with adverse outcome in COVID-19 patients and might be useful in the treatment of viral infections by preventing or ameliorating T cell exhaustion. Authors described future directions in this field.

Comments:

  1. Introduction and mode of action of ICI sessions might be shortened because they were well known facts. This manuscript is for “Perspective”.
  2. Immune reconstitution syndrome was originally reported to occur in the setting of HIV infection. The syndrome might occur in ICI-treated patients who were infected with COVID-19 because ICI promote T cell activation. Please discuss the point. Two reports from Allergology International (2010;59:333-343) and Thoracic Cancer (2020; 11: 1330-1333) might be useful.
  3. P7, line 332: on their recommendations. many cancers -> on their recommendations, many cancers

Author Response

We are very grateful for the reviewer comments. We have replied to their comments point by point in italics:

Ref. 2

Comments and Suggestions for Authors

ICI can promote the exacerbated immune response associated with adverse outcome in COVID-19 patients and might be useful in the treatment of viral infections by preventing or ameliorating T cell exhaustion. Authors described future directions in this field.

Comments:

  1. Introduction and mode of action of ICI sessions might be shortened because they were well known facts. This manuscript is for “Perspective”.

We shortened these sections as requested

  1. Immune reconstitution syndrome was originally reported to occur in the setting of HIV infection. The syndrome might occur in ICI-treated patients who were infected with COVID-19 because ICI promote T cell activation. Please discuss the point. Two reports from Allergology International (2010;59:333-343) and Thoracic Cancer (2020; 11: 1330-1333) might be useful.

We included  the references and briefly discussed this point in section 5.

Shiohara T, Kurata M, Mizukawa Y, Kano Y. Recognition of immune reconstitution syndrome necessary for better management of patients with severe drug eruptions and those under immunosuppressive therapy. Allergol Int. 2010 Dec;59(4):333-43. doi: 10.2332/allergolint.10-RAI-0260. Epub 2010 Oct 25. PMID: 20962568.

Gozzi E, Rossi L, Angelini F, Leoni V, Trenta P, Cimino G, Tomao S. Herpes zoster granulomatous dermatitis in metastatic lung cancer treated with nivolumab: A case report. Thorac Cancer. 2020 May;11(5):1330-1333. doi: 10.1111/1759-7714.13377. Epub 2020 Mar 5. PMID: 32141197; PMCID: PMC7180604.

Cancio M, Ciccocioppo R, Rocco PRM, Levine BL, Bronte V, Bollard CM, Weiss D, Boelens JJ, Hanley PJ. Emerging trends in COVID-19 treatment: learning from inflammatory conditions associated with cellular therapies. Cytotherapy. 2020 Sep;22(9):474-481. doi: 10.1016/j.jcyt.2020.04.100. Epub 2020 May 7. PMID: 32565132; PMCID: PMC7252029.

  1. P7, line 332: on their recommendations. many cancers -> on their recommendations, many cancers

We have corrected this typo.